# Classification of *Prunus* Genus by Botanical Origin and Harvest Year Based on Carbohydrates Profile

**DOI:** 10.3390/foods11182838

**Published:** 2022-09-14

**Authors:** Marius Gheorghe Miricioiu, Roxana Elena Ionete, Diana Costinel, Oana Romina Botoran

**Affiliations:** 1ICSI Analytics Group, National Research and Development Institute for Cryogenics and Isotopic Technologies—ICSI, 240050 Râmnicu Vâlcea, Romania; 2Academy of Romanian Scientists, Splaiul Independentei 54, 050094 Bucharest, Romania

**Keywords:** ^1^H-NMR, carbohydrates, fruits, PCA, LDA

## Abstract

The ^1^H-NMR carbohydrates profiling was used to discriminate fruits from *Rosaceae* family in terms of botanical origin and harvest year. The classification was possible by application of multivariate data analysis, such as principal component analysis (PCA), linear discriminant analysis (LDA) and Pearson analysis. Prior, a heat map was created based on ^1^H-NMR signals which offered an overview of the content of individual carbohydrates in plum, apricot, cherry and sour cherry, highlighting the similarities. Although, the PCA results were almost satisfactory, based only on carbohydrates signals, the LDA reached 94.39% and 100% classification of fruits according to their botanical origin and growing season, respectively. Additionally, a potential association with the relevant climatic data was explored by applying the Pearson analysis. These findings are intended to create an efficient NMR-based solution capable of differentiating fruit juices based on their basic sugar profile.

## 1. Introduction

Nowadays, people pay particular attention to a balanced and controlled diet. Thus, there is a high demand for fresh fruits and derived juices which are considered, along with vegetables, the healthiest foods. People have also begun to recognize their beneficial contribution to health by protecting the human body against different type of illnesses through their vitamins which usually result in an immunity increase [1]. Unfortunately, this high demand for quality fruit juices has also attracted some frauds as shown in several studies on this topic [2,3,4,5]. Traceability, establishing chemical profile, appropriate physical attributes, adequate textural properties, controlled toxins and microbial contamination, as well as processing and storing method all represent characteristics of high quality, particularly for agricultural products. For example, a common fraudulent act is represented by the adulteration of juices by addition of other types of cheaper juices obtained from less expensive or more common fruits in the respective area [6]. Among others, dilution with water, sugar syrup and colorants, production method (conventional, organic, traditional techniques) as well as non-declared processing technologies (freezing, irradiation) can be mentioned [7]. Fruit juices have a high added value, and they are also more vulnerable to being subjected to different adulteration techniques because of their acknowledged attributes. Accordingly, authorities must be able to determine the compliance of a suspect product based on the product description, identify fraudulent processing practices, prevent adulteration, and control any other practices that may deceive the consumer. As is well known, it is not only consumers who benefit from food authenticity assessment but also the food industry parties which rely on the ability to ensure their commodities’ label conformity and brand protection. In order to verify the authenticity of food items and guide and assist law enforcement, official bodies are periodically requesting an updated list of analytical procedures. In this scope, many analytical techniques such as molecular techniques (DNA-based approaches) [8], isotopic approaches [9], ultraviolet-visible spectrophotometry (UV-VIS), high performance liquid chromatography (HPLC), gas chromatography (GC), inductively coupled plasma mass spectrometry (ICP-MS), atomic absorption spectroscopy (AAS), and infrared spectroscopy (IR) are used to control and to detect the food products’ adulteration [2,6]. A single analyte or approach may rarely be associated with overall quality compliance because of the fruit juice matrices’ complexity. As a result, their quality is derived from a unique combination of characteristics. To acquire the defined quality markers and control the critical production parameters, it is often essential to use multivariate data analysis. It is easier to discriminate between fruit samples and establish their authenticity when multivariate analysis is used in conjunction with different spectroscopic or chromatographic based methods. Data fusion may offer more precise information about a sample and better interpretation than a single approach, but usually the use of several techniques is not economically profitable, doesn’t respect green chemistry principles, and is time consuming. In this respect, nuclear magnetic resonance (NMR) is a non-destructive technique providing high analytical precision, enabling simultaneously compounds identification, exposing complex frauds, and, in conjunction with chemometric analysis, revealing possible markers of fruits authenticity based on their composition profile [10]. Moreover, through NMR, spectra fruits can be classified according to their varietal and geographical origin [11,12,13,14], this being possible by application of some dedicated instruments, such as unsupervised machine learning techniques (principal component analysis—PCA) and pattern-recognition tools (discriminant analysis—DA).

The ^1^H-NMR spectra in combination with PCA and DA have been applied to evaluate changes in the composition and metabolic profile of juices during thermal concentration process [15] for the correlation of different varieties of fruits [16,17] to reveal the juices’ adulteration [18], and to evaluate the plant growth regulator in strawberries [19]. Also, the metabolomic analysis of ^1^H NMR results obtained from fruit juices investigation give an overview about the relationships between the major metabolites and the sensory characteristics of the fruits. Among these metabolites, the carbohydrate content is the most relevant for the maturity level of fruits and for the consumer perception, and its domain is represented by three predominant components: glucose, fructose and sucrose [20,21]. Generally, the glucose and fructose are presented in lower quantities than sucrose and their relation affects the taste of the fruits [22]. Thus, the sweetness of the fruits is directly influenced by the fructose quantity which is 2.3 and 1.7 times sweeter than glucose and sucrose, respectively [20]. Beside this, fructose is highly appreciated for its therapeutic effect on the gastrointestinal tract [22,23]. Also, sucrose is appreciated as sweetener, energy source, and antioxidant [22], but its excessive consumption is closely related to the risk of caries, obesity, and diabetes [24].

In this study, the NMR method coupled with multivariate statistical analysis was used to obtain information about the variation of carbohydrate content in different fruits (plum, cherry, sour cherry, and apricot) from the *Rosaceae* family, *Prunus* genus, harvested in three different years. Furthermore, discrimination models based on DA were developed for samples classification. Another approach was the evaluation of different climatic conditions’ (temperature and precipitation) influence over the three investigated years on the fructose, sucrose, glucose, and total carbohydrates content. These data will represent a contribution to regional horticultural varieties’ characterization as well as provide useful information for industries which use fruit-derived nutrients in food production.

## 2. Materials and Methods

### 2.1. Chemicals

All analytical standards and reactants used for samples preparation and data interpretation were purchased from Sigma-Aldrich (St. Louis, MO, USA) and are hereafter listed: HCl (5N), NaOH (5N), D2O, and TMSP.

### 2.2. Sample Collection and Pre-Treatment

The fruits analyzed in this study were chosen from the *Rosaceae* family, *Prunus* genus, namely plum, cherry, sour cherry, and apricot. In particular, 76 samples were analyzed from Romania. The fruits were provided by Vâlcea Fruit Growing Research and Development Station, Romania, harvested in a state of consumption maturity, and the collected fruit samples were cooled and transported to the laboratory, assuring the maintenance of the cold chain. Then, the fruits (approximately 5 kg per variety) were washed with water, kept frozen, and stored at −20 °C in a freezer until sample preparation. 

The juice was obtained by squeezing the whole fruit using a juicer (Moulinex, Jinan, China), and then approximately 100 g of sample was centrifuged with 10,000 rotations/min (Hettich ROTINA 420, Tuttlingen, Germany) for 10 min and filtered through a filter with 45 µm porosity, leading to about 20 mL being obtained. Further, the samples were pH adjusted to 2.65 by using 5 N HCl and 5 N NaOH. For NMR analysis, 700 μL of each sample was combined with 70 μL of deuterium oxide (D_2_O), 99.9% D containing, and 0.05 wt. % of 3-(trimethylsilyl) propionic-2,2,3, 3-d 4 acid sodium salt as an internal standard (TMSP). The mixtures were transferred to 5 mm NMR tubes. 

### 2.3. NMR Analysis and Data Processing

All ^1^H-NMR spectra were recorded at 300 K temperature on a 400 MHz Bruker Avance spectrometer (Bruker France SAS, Wissembourg, France), operating at 9.4 T, equipped with a 5 mm BBO probe and ATM (Automatic Tuning Matching). In addition, the instrument was fitted with an autosampler from Bruker controlled by Icon NMR software which allows a loading of 60 samples. To complete the temperature equilibration, a time delay of 5 min between sample injection and preacquisition calibrations was set. The suppression of H_2_O signals was assured through Bruker standard pulses sequence, noesygppr-1d, by applying continuous waves during the relaxation delay (10 s) with a mixing time of 10 ms. Each spectrum is the result of 8 scans and 32768 (33 k) data points. The spectral width was adjusted to 6402 Hz with an acquisition time of 2.559 s per scan. Spectra were Fourier transformed, manually phased, baseline corrected, and referenced to TMSP signal at 0 ppm using TopSpin 3.2 software (Bruker Biospin, Rheinstetten, Germany). Principal component analysis (PCA) and discriminant analysis (DA) were performed with LSTAT Addinsoft 2014.5.03 software version (Addinsoft, New York, NY, USA) in order to evaluate some potential variables and their influence on fruit juices’ discrimination. Also, to reveal the possible relationships between sugars and climatic conditions, the Pearson correlations coefficient at *p* ≤ 0.05 was used.

## 3. Results and Discussion

### 3.1. Fruits ^1^H-NMR Spectra and Assignment of the Interest Peaks

In order to investigate the potential classification of fruit juices according to their botanical origin (apricot, cherry, sour cherry, and plum) and harvest year (2015, 2016, and 2017) their metabolic profile was obtained by using the ^1^H-NMR spectroscopy. A typical 400 MHz ^1^H 1D-NOESY NMR spectrum for a juice sample is shown in Figure 1. 

Generally, the fruit juices’ spectra are dominated by α-glucose, β-glucose, sucrose, and fructose signals, having the highest concentration among all the other metabolites. These belong to the carbohydrates region of the spectrum, which is placed in the middle-frequency region of the spectrum between 3.0 and 6.0 ppm, and it is followed by, in terms of signal intensities, amino acids, aliphatic region (from 0.5 and 3.0 ppm), and phenolic region (from 6.0 to 8.5 ppm) [16,17,25,26,27,28,29]. When the 5.40 to 4.10 ppm region was investigated, a pair of duplets were observed at 5.22 and 4.63 ppm that correspond to α and β glucose hydrogen in position 1. In the same range was identified one more pair of duplets at 5.39 and 4.20 ppm, which were assigned to hydrogen in position 1 from glucose and position 2 from fructose found in the composition of a sucrose molecule. Among the investigated spectra, sucrose signals were also detected at 5.40, from 4.19 to 4.21 ppm, 3.75 to 3.85 ppm, 3.67 and 3.54 ppm, while glucose signals were observed at 5.23, 5.22, 3.76, 3.71, 3.51, 3.505, 3.43, and 3.41 ppm, and from 4.64 to 3.21 ppm. Fructose signals were in the range of 3.57 to 3.60, 4.10 to 3.99 ppm, and 3.68 to 3.77 ppm, while the peaks were found at 3.60 and 4.10 ppm. The identification of carbohydrates signals was possible by consulting the literature [15,18,25,26,27,28,29]. Thus, the relevant ^1^H-NMR peaks obtained for the carbohydrates region are reported in Table 1, and, for fruit juices’ classification, each signal was taken into account, this region summing a total of 38 signals. These signals were different for all the studied fruit juices. For example, sucrose showed higher intensities in plum and apricot juices, while α-glucose, β-glucose, and fructose signals were more intense in the case of cherry and sour cherry juices.

### 3.2. Fruits Variety-Based Classification

In order to investigate the similarities between carbohydrates’ metabolites in different fruit varieties, a heat map was generated (Figure 2). The heat map was built on the signals of sucrose, fructose, α-glucose, and β-glucose. As it can be seen from Figure 2, plums and sour cherries have the same content of sucrose, while cherry and sour cherry, as in the case of plum and apricot, have the same content of fructose. Moreover, similar contents of α-glucose and β-glucose in cherry and apricot compositions can be observed. The lower level of sucrose than α-glucose and β-glucose in cherry juices was also indicated in other study [28], this sugar being produced only in leaves by photosynthesis and translocated from different parts of the tree through phloem [20]. Therefore, the sucrose content is directly proportional with the photosynthetic rate. Beside this, the changes in fruit metabolism and the dilution caused by the fruit volume increase have significant effect on the whole carbohydrates content [20].

As previously stated, the NMR profiles reveal the existence of the same compounds but in different quantities. In order to determine the existence of latent variables linking different compounds, principal component analysis was performed to the intensity of 38 ^1^H resonances (Table 1 and Figure 3). PCA analysis provided additional information regarding the separation of fruit juices, in term of botanical origin, as well as variable reduction, and assessment of clustering in PCA score. The four total principal components extracted, whose eigenvalues exceeded 1, explained 84.86% of the total variance. From this cumulative percentage the first two principal components accounted for 78.61% (72.81% for PC1 and 5.80% for PC2) indicating that it can be applied to obtain sample clusters in two-dimensional space. Thus, Figure 3 revealed a slight separation between the fruits with different botanical origin. The clearest separation is between apricots and the other fruit juices. The main differentiation was performed among the first principal component. Sour cherry and cherry juices samples have negative values for PC1, while apricot samples have positive values for PC1. A visible trend of separation is observed between cherries and sour cherries among PC2, the last being positively correlated with PC2. The plum samples are scattered, without a clear separation tendency.

For a closer examination regarding the relationships between the fruit variety and certain metabolites, the high dimensional data was projected to a lower dimensional subspace by means of the calculated principal components. (Figure 3b). Thus, it can be observed that the apricot juice is defined by a high content of sucrose, and cherry and sour cherry juices present a higher content of fructose, α-glucose, and β-glucose, while plums are characterized by a moderate content of sugars. Sugar accumulation in fruit is a dynamic quantitative trait that is usually influenced by environmental factors and is based on a variety of related physiological and biochemical processes. It is also determined by a number of enzymes that are correlated with the natural ecosystem and agricultural practices. Sucrose is generally accepted as the predominant sugar present in apricot fruit followed by glucose and fructose [30,31]. The mechanisms that influence the sugar profile of fruits have not been thoroughly studied. In apricots from several apricot-growing regions around the world, the proportions of the four carbohydrates varied substantially (sucrose: 18 to 82%; glucose: 5 to 28%; fructose: 2 to 17%). Individual sugar levels may vary significantly as a consequence of both genetic variability and environmental factors. According to Zhang [32], the patterns of each carbohydrate in apricots are genetically controlled, the accumulation being correlated to the sucrose-metabolizing enzyme activity [32]. Another important aspect regarding apricot sucrose content is related to the harvest maturity. Generally, the growth pattern of stone fruits is presented as a double-sigmoidal curve with three distinct growth phases: stage I characterized by a rapid growth period, stage II presented as a period of reduced growth, and stage III characterized by an even more rapid growth [33]. Overall, in a study conducted by Xi [34] regarding apricot fruit development and ripening, it was discovered that all sugars present an amplified accumulation pattern during the whole growth phase, with glucose and sucrose being the predominant identified carbohydrates. During the first stage, glucose was the most abundant sugar, but, as the process advanced, sucrose concentration increased exponentially. At the end of stage III, the sucrose concentration surpassed the total amount of glucose [34]. The investigation performed by Bae [35] presented similar results, namely sucrose content being smaller or even undetectable compared to glucose and fructose in the first stage of growth, followed by a significant increase at the full maturation stage [35]. These results indicate that the apricot carbohydrates accumulation metabolism shifts from glucose-predominated to sucrose-predominated during fruit development and ripening, presenting a balanced transition from synthesis to degradation. Furthermore, harvest maturity assessment represents a key component in determining fruit quality and customer acceptance [36]. Despite the fact that the apricot harvest ripeness period depends on the intended application, such as fresh consumption in local markets or long-term transit, realizing it too soon may have a detrimental impact among fruit sensory quality (as they will not be able to continue maturing or ripen properly), even if it makes it more resistant to postharvest handling. Apricots picked at appropriate maturity indexes are expected to present a higher customer satisfaction than those harvested at commercial ripe maturity, as is the case of peaches or nectarine [37]. All of these findings suggest that apricot sucrose content could be considered a variety and maturity marker. As a result, the PCA scores plot represented a valuable tool for visualizing possible discrimination within the data set that can almost classify the juices, from Prunus genus but does not represent a perfect analysis to complete the botanical origin separation based only on carbohydrates data. In this regard, for further investigation of potential botanical origin separation of fruit juices, LDA is a technique that maximizes group separation (Figure 4). LDA proceeds by constructing discrimination functions from linear equations of variable data sets. The obtained model could be used to classify unknown observations (such as questionable or unknown samples). LDA was generated directly to the raw data set even if spectral data are known to be highly collinear.

Three discriminant functions were obtained by LDA, which were demonstrated as suitable for correct classification. A total of 100% of the distribution was explained by this model; the first discriminant function accounted for 74.93%, the second function for about 19.46%, and the third one for 5.61%. Analyzing Figure 4a, it can be observed that the plum samples are found in the superior quadrant. The apricot samples are distributed in the third quadrant, excepting one sample. For the sour cherry and cherry, the results are not satisfactory because, as in the case of the other fruits, the separation among the two groups was not achieved due to some observations that overlapped. Function 1 provides the main separation between apricots and the other three fruit juices and was primarily correlated with sucrose, followed by fructose. Generally, the LDA results were superior and a reliable classification of fruit juices by botanical origin was achieved, except the cherry and sour cherry juices, where a slight separation was observed when using all three discriminant functions. The third discriminant function was mainly correlated with the sucrose signals, these not being identified in the cherries’ and sour cherries’ spectrum or being present in very low concentrations. The apricot, cherry, plum, and sour cherry juices were correctly classified with 100%, 100%, 96.30%, and 91.67%, respectively. The results obtained by the two applied multivariate techniques are similar; this fact implies that the results are reliable. More than 92% of the total fruit juices are classified correctly. However, these results could be biased to a certain extent, due to the unbalanced number of samples in each class; there two times more plum samples than apricot and sour cherry. Despite this drawback, the classifications for apricot, cherry, and plum are promising, once the classes are equilibrated.

### 3.3. Harvest Year-Based Classification and Climatic Condition Influence

For harvest year-based classification of fruit juices, the same procedure was followed, but the signals of carbohydrates were grouped after the three growing years (2015, 2016, and 2017) in order to obtain the statistical analysis. 

The PCA results according to growing year are shown in Figure 5a. Unfortunately, the plots are spread over the all quadrants and no clusters were formed based on the harvest years.

Despite the fact that PCA presented a differentiation between the botanical origin of the fruits, in the case of the harvest years, the discrimination within each other with the same method was not achieved. In this respect, to come with a supplement to strengthen the results or to clear up some ambiguities related to PCA analysis, the same data were subjected to LDA analysis. The results of the LDA are superior (Figure 5b), and, according to the confusion matrix for the cross-validation results, a 96.05% was reached (two 2015 samples were classified as 2017, and one 2017 sample was classified as 2015). The first function accounted for 75.33% and the second function for 24.67%. All three groups of juice samples from fruits harvested in 2015, 2016, and 2017 are visibly separated. Taking into account the different botanical origin of the samples, a good classification of the three harvest years was done.

Due to the fact that each growing season could be different from the point of view of temperatures and precipitation, the potential influence of climatic condition on the sugar content was accessed by applying the Pearson analysis. Prior, the percentage of individual sugar from total sugars and the average temperature and precipitation recorded in the growing months of each season (June, July and August) for the studied years were calculated.

According to climatic condition data (temperature and precipitation), the growing seasons were different, presenting for the 2015 harvest year a seasonal average temperature of 20.8 °C and 58.3 mm precipitation, 20.3 °C and 86.7 mm for 2016, and a higher temperature (22.5 °C) and lower precipitation (66.7 mm) for 2017. In 2016, the average precipitation quantities were significantly higher than in 2015 and 2017, while the same year recorded the lowest average temperature, with almost 2 degrees below the average temperature recorded in 2017.

As it is shown in Table 2, the fructose and β-glucose are negatively correlated with temperature and positively with precipitation. Unfortunately, correlation between sucrose and precipitation or temperature was not found. From the same analysis, it can be remarked that fructose is highly and positively correlated with α-glucose and β-glucose. Instead, the sucrose contents significantly and negatively correlated with fructose, α-glucose, and β-glucose. Fruit juice quality is generally controlled by the growth conditions, being first dependent on the growing region climate (terroir, rainfall, humidity, hours of sun, temperature day/night, etc.) and secondly on technological influences. If the vintage year doesn’t provide ideal circumstances for grape growth and quality, technology and enology can only have a limited impact. Even if a strong correlation was not observed, the temperature affects seedling development, photosynthesis, and the soluble sugar (fructose, glucose, and sucrose) content during fruit production [38,39]. The sugar content in various plant parts is known to decrease at high temperature due to plants’ defense mechanisms to use photosynthetic products to support higher metabolic activities, while lower temperatures encouraged sugar accumulation. The observed differences in temperatures could have influenced the investigated fruits’ soluble sugar leading to a possible explanation of the separation of the fruits according to the harvest.

## 4. Conclusions

Untargeted ^1^H-NMR carbohydrates profiling of fruit juice proved to be a powerful tool for the classification and characterization of different *Prunus* sp. varieties. Apricot, cherry, sour cherry, and plum juices’ ^1^H-NMR spectra were observed to be dominated by sugars, which play a key role in defining the fruit’s taste and flavor and make them good phytomarkers for species differentiation. Both botanical origin and harvest year could be assessed by means of multivariate analysis of the data, highlighting fruit specific sugar chemical traits. The carbohydrates profile differs among the analyzed fruit juices, and, based on their signal intensities, the PCA revealed strong correlation between sucrose and apricot juice, while the fructose and glucose were correlated with cherry and sour cherry juices. Despite these specific correlations, for a better botanical origin classification, an LDA was performed. A successful two-dimensional separation between plum and apricot juices was achieved, whereas cherry and sour cherry juices were slightly differentiated by implying all three discriminant functions. Additionally, the LDA represents a powerful technique for harvest year separation as the different botanical origin juices with different growing seasons (2015, 2016 and 2017) were clearly separated. From the point of view of climatic condition, by applying Pearson analysis, some correlations between fructose and β-glucose with temperature and precipitation were noted. However, solid associations between basic climatic factors and certain carbohydrate metabolic profiles could not be found, further studies being needed in order to obtain better correlations. Nevertheless, these results suggest the possible use of NMR-based sugar profiling for *Prunus* sp. botanical origin prediction and assessment of the possible correlation with different fruit juices, and more models can be developed for future predictions related to their quality and authenticity. In this regard, it should be noted that quality assessment was not the primary goal of the current fruit juices carbohydrate profile characterization research, which simply aimed to indicate potential metabolic profile variations across different species.

## Figures and Tables

**Figure 1 foods-11-02838-f001:**
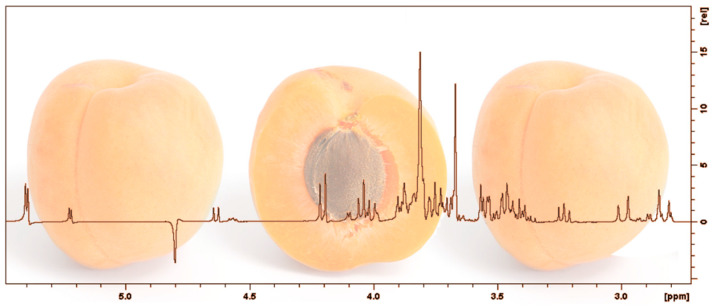
General ^1^H-NMR spectra of apricot fruit juice—carbohydrates region.

**Figure 2 foods-11-02838-f002:**
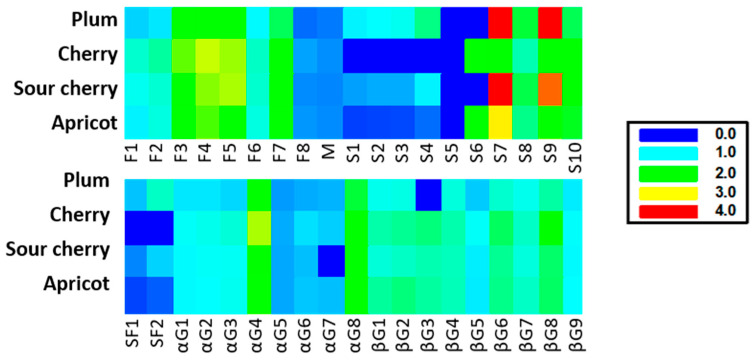
Heat map built on ^1^H-NMR carbohydrates signals (F1–F8, fructose signals; M, methanol signal; S1–S10, sucrose signals; SF1 and SF2, sucrose-fructose signals; αG1–αG8, alpha glucose signals; and βG1–βG9 represent the beta glucose signals).

**Figure 3 foods-11-02838-f003:**
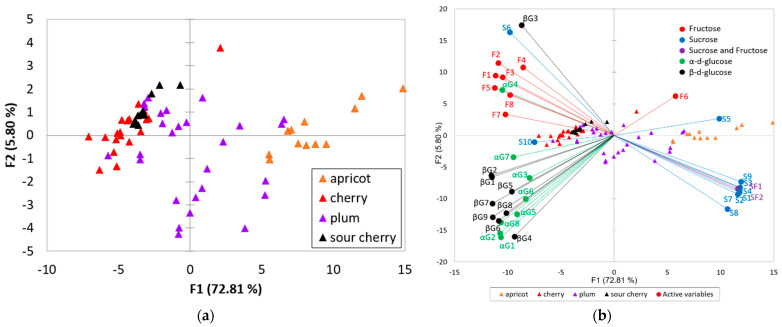
(**a**) PCA score plot of four fruit samples’ variety derived from conventional ^1^H-NMR spectra; (**b**) correlation between the signals and factors responsible for fruits’ variety separation.

**Figure 4 foods-11-02838-f004:**
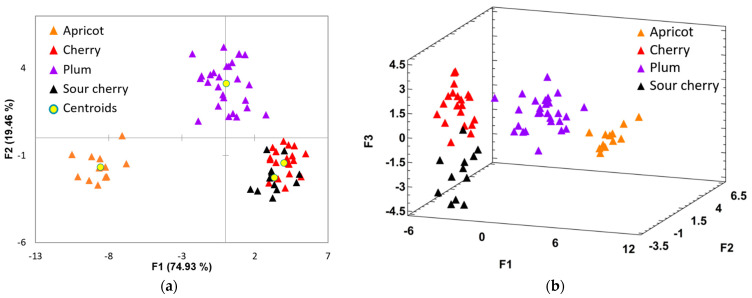
(**a**) 2D and (**b**) 3D plots showing the discrimination of fruit juices samples according to their botanical origin.

**Figure 5 foods-11-02838-f005:**
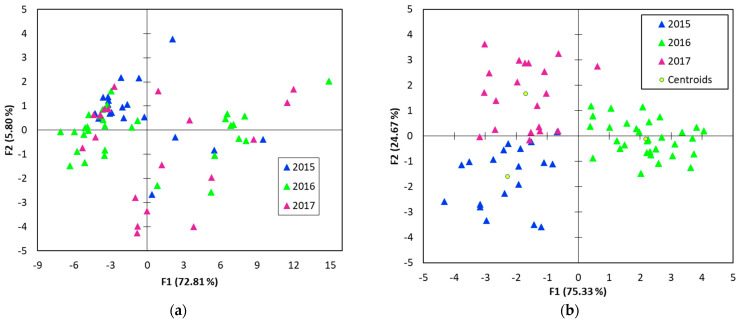
(**a**) PCA and (**b**) DA F1/F2 score plot showing the separation between the 3 harvest years.

**Table 1 foods-11-02838-t001:** Chemical shift (δ) and assignment of metabolite resonances in the ^1^H-NMR spectra of plum, cherry, sour cherry, and apricot juices.

Metabolites	δ (ppm), Multiplicity (*j*, Hz) and Assignment			
	Plum	Cherry	Apricot	Sour Cherry
	*Prunus domestica*	*Prunus avium*	*Prunus armeniaca*	*Prunus cerasus*
	(*n* = 27)	(*n* = 22)	(*n* = 15)	(*n* = 12)
β-d-glucose	3.23 (dd, CH),	3.23 (dd, CH),	3.23 (dd, CH),	3.23 (dd, CH),
	3.40 (dd, CH),	3.40 (dd, CH),	3.40 (dd, CH),	3.40 (dd, CH),
	4.63 (d, H1)	4.63 (d, H1)	4.63 (d, H1)	4.63 (d, H1)
methanol	3.36 (s, CH3)	3.36 (s, CH3)	3.36 (s, CH3)	3.36 (s, CH3)
α-d-glucose	3.43 (dd, CH),	3.43 (dd, CH),	3.43 (dd, CH),	3.43 (dd, CH),
	3.50 (dd, CH),	3.50 (dd, CH),	3.50 (dd, CH),	3.50 (dd, CH),
	5.22 (d, CH)	5.22 (d, CH)	5.22 (d, CH)	5.22 (d, CH)
fructose	3.60 (d, CH2)	3.60 (d, CH2)	3.60 (d, CH2)	3.60 (d, CH2)
	3.99 (H5),	3.99 (H5),	3.99 (H5),	3.99 (H5),
	4.10 (d, H3, H4)	4.10 (d, H3, H4)	4.10 (d, H3, H4)	4.10 (d, H3, H4)
sucrose	4.20 (d, H3),		4.20 (d, H3),	
	5.39 (d, H1)		5.39 (d, H1)	

**Table 2 foods-11-02838-t002:** Pearson correlation coefficients between precipitation, temperature and carbohydrates.

Variables	Temperature	Precipitation	Carbohydrates	F	S	SF	αG	βG
Temperature	**1**	−0.564	−0.047	−0.261	0.121	0.105	−0.066	−0.173
Precipitation	−0.564	**1**	0.089	0.107	0.089	0.092	−0.036	0.083
Carbohydrates	−0.047	0.089	**1**	0.567	−0.672	−0.679	0.693	0.698
F	−0.261	0.107	0.567	**1**	−0.856	−0.863	0.778	0.816
S	0.121	0.089	−0.672	−0.856	**1**	0.993	−0.908	−0.885
SF	0.105	0.092	−0.679	−0.863	0.993	**1**	−0.893	−0.886
αG	−0.066	−0.036	0.693	0.778	−0.908	−0.893	**1**	0.903
βG	−0.173	0.083	0.698	0.816	−0.885	−0.886	0.903	**1**

## Data Availability

The datasets generated for this study are available on request to the corresponding author.

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
