# Peer review of "Classification of Prunus Genus by Botanical Origin and Harvest Year Based on Carbohydrates Profile"

_foods, 2022, doi:10.3390/foods11182838_

Round 1
Reviewer 1 Report
The study applied 1H NMR for juice classifying by sugar contents. The data is solid. Some commends are provided to the author.
1. Line 42, a “c” may be removed
2. Line 64-67. Please provide reference.
3. Many articles in literature have dressed using H NMR of sugar profile for fruit adulteration such as cherries. Please add the novelty and the difference of your paper to other papers. eg. Non-targeted 1H NMR fingerprinting and multivariate statistical analyses for the characterisation of the geographical origin of Italian sweet cherries. 2013. Food Chemistry, 141(3), 3028-3033.
4. Line 108. The concentrations of HCl and NaOH may be addressed, because proton may be added into sample.
5. Figure 2 is suggested to added the definition of F1-F8, M, S1-S10, SF1-SF2, alphaG1-betaG9.
6. For PCA analysis, since there is no data of sugar contents present in the paper. It is suggested to put some data of sugar contents in the sample in this paper. To dress how different of the sugar contents among samples by real data.
7. There is no correlation among Temperature, precipitation, and carbohydrates, only sugars are correlated. Why author introduce temperature and precipitation into the discussion? And what is the finding of the relation among temperature, precipitation and sugars?
8. In conclusion, the PCA method only classified pure juice. What is the result if the juice was adulterated with 50% or less other juice?
Author Response
Dear reviewer,
First of all, thank you for the professional comments and observations regarding the paper “Classification of Prunus genus by botanical origin and harvest year based on carbohydrates profile” by Marius Gheorghe Miricioiu, Oana Romina Botoran, Diana Costinel, Raluca Popescu, and Roxana Elena Ionete. We thank for comments, which have made us think carefully about our data sets again. Accordingly, we have reanalyzed these where necessary. Please find below our point-by-point itemized answer and correction. We write to say that we now strongly believe that we can convince you that the data is sound and that we have adequately answered in various valid concerns.
The study applied 1H NMR for juice classifying by sugar contents. The data is solid. Some commends are provided to the author.
- Line 42, a “c” may be removed
Author’s response: We have removed the “c” letter.
- Line 64-67. Please provide reference.
Author’s response: The references [11-14] were introduced for the comment between the lines 64-67.
- Many articles in literature have dressed using H NMR of sugar profile for fruit adulteration such as cherries. Please add the novelty and the difference of your paper to other papers. eg. Non-targeted 1H NMR fingerprinting and multivariate statistical analyses for the characterization of the geographical origin of Italian sweet cherries. 2013. Food Chemistry, 141(3), 3028-3033.
Author’s response: Our paper aim was to highlight the principal differences existing in the sugar profiles of 4 different fruit juice originating from the same genus but with different economic values, especially for the Romanian market. The identified carbohydrates could be used as origin markers to a specific variety, thus being useful for adulterated fruit juices detection especially in cases of fraud by substitution. By comparing our paper novelty with the suggested article two different directions of applicability of the NMR technique can be observed, our, that address the (i) botanical origin and harvest year and (ii) the geographical origin, these works, as well as many others, demonstrating NMR as a decision-making tool, and at the same time the need for development and implementation of profile techniques in food authenticity as official control methods.
- Line 108. The concentrations of HCl and NaOH may be addressed, because proton may be added into sample.
Author’s response: The HCl and NAOH concentration were introduced in the paper.
- Figure 2 is suggested to added the definition of F1-F8, M, S1-S10, SF1-SF2, alphaG1-betaG9.
Author’s response: In the picture description we introduce the definitions for the used abbreviations.
- For PCA analysis, since there is no data of sugar contents present in the paper. It is suggested to put some data of sugar contents in the sample in this paper. To dress how different of the sugar contents among samples by real data.
Author’s response: For the PCA analysis were used the different signal intensities that are direct proportional with the carbohydrate’s concentration as a semiquantitative analysis. The principal aim of this paper was to evaluate if by using the entire sugar profile, without quantifying the sugar content, is suitable as a decisional instrument for species identification. This could be used as a future direction in order to detect adulterated mixtures or frauds such substitution of high-quality raw material with a cheaper one. So, by identifying specific compounds that characterized a juice, could be use as a marker for different mixtures.
- There is no correlation among Temperature, precipitation, and carbohydrates, only sugars are correlated. Why author introduce temperature and precipitation into the discussion? And what is the finding of the relation among temperature, precipitation and sugars?
Author’s response: We tried to identify if there were correlations among the sugar profile and temperature and precipitation because fruit juices quality is generally controlled by the growth conditions, being first dependent on the growing region climate (terroir, rainfall, humidity, hours of sun, temperature day/night, etc.). Because we previously investigated harvest year-based classification, we thought that it will be important to check if there was a direct correlation between the temperature/precipitation during maturation stage and fruits sugars, since they can be inhibited or not by these characteristics.
- In conclusion, the PCA method only classified pure juice. What is the result if the juice was adulterated with 50% or less other juice?
Author’s response: The NMR technique can be used as a first step in the samples authenticity investigations of the carbohydrate profile providing useful information for classifying suspect, adulterated or authentic, always by comparing with a reference, hence the need to build authentic databases, specific to each country and fruit species. For example, if a peach juice will be adulterated with a cheaper juice such as apple, by doing a simple sugar NMR profile we could identify a specific compound, such as α- or β-Galactose (typical sugar for apples).
Thank you very much!
Best regards,
Oana BOTORAN

Reviewer 2 Report
Dear appreciated Authors,
The paper represents an actual and great contribution to science. While quality of the paper is very good.
1. Line No. 23: Nowadays, the peoples pay particular attention to balanced and controlled diet. (Instead close attention)
Line 34: Among others, it can be also mentioned the dilution with
Table 1. Please prepare Table 1. according to Guidelines
Line 304 - 309: I suggest to present Climatic data in one Table to be more clear if it is possible.

Author Response
Dear reviewer,
First of all, thank you for the professional comments and observations regarding the paper “Classification of Prunus genus by botanical origin and harvest year based on carbohydrates profile” by Marius Gheorghe Miricioiu, Oana Romina Botoran, Diana Costinel, Raluca Popescu, and Roxana Elena Ionete. We thank for comments, which have made us think carefully about our data sets again. Accordingly, we have reanalyzed these where necessary. Please find below our point-by-point itemized answer and correction. We write to say that we now strongly believe that we can convince you that the data is sound and that we have adequately answered in various valid concerns.
- Line No. 23: Nowadays, the peoples pay particular attention to balanced and controlled diet. (Instead, close attention)
Author’s response: The word “close” was replaced with “particular”.
- Line 34: Among others, it can be also mentioned the dilution with
Author’s response: We modify as suggested.
- Table 1. Please prepare Table 1. according to Guidelines
Author’s response: We changed the table design according to the paper Guidelines
- Line 304 - 309: I suggest to present Climatic data in one Table to be more clear if it is possible.
Author’s response: We tried to make a table as suggested but due to the fact that are only six values to introduce for 3 observations, we would like to maintain the data in text, thank you for understanding.
Thank you very much!
Best regards,
Oana BOTORAN

Reviewer 3 Report
1. Abstract: Mention the number of samples (calibration and prediction) used in this study
2. Introduction: Please mention the reason why using the NMR method since this method is more expensive than other spectroscopic methods (for example UV-vis).
3. Sample collection: How did the authors select the sample with similar maturity stages? Then 100 grams from each fruit were used. How did the authors collect the 100-gram samples? From the same part of each fruit or from several parts of the fruits? Please mention it in more detail.
4. Figure 4. Please revise. Remove the 2D plot (left part). It is not in line with the explanation in the text.
5. Figure 5 (b). Some samples overlapped. How did you calculate 100% correct classification? Show the result of the classification in detail.
6. The authors mention the use of PCA and LDA. The two are different in the number of samples used in the calculation. For LDA, how did the authors separate the samples (for calibration and prediction)?
Author Response
Dear reviewer,
First of all, thank you for the professional comments and observations regarding the paper “Classification of Prunus genus by botanical origin and harvest year based on carbohydrates profile” by Marius Gheorghe Miricioiu, Oana Romina Botoran, Diana Costinel, Raluca Popescu, and Roxana Elena Ionete. We thank for comments, which have made us think carefully about our data sets again. Accordingly, we have reanalyzed these where necessary. Please find below our point-by-point itemized answer and correction. We write to say that we now strongly believe that we can convince you that the data is sound and that we have adequately answered in various valid concerns.
- Abstract: Mention the number of samples (calibration and prediction) used in this study
Author’s response: In order to build the classification models, the samples were split randomly into the calibration set and the prediction set at a ratio of 2:1.
- Introduction: Please mention the reason why using the NMR method since this method is more expensive than other spectroscopic methods (for example UV-vis).
Author’s response: Comparing the two methods, the NMR provide a full profile spectrum of the components present in the analyte. Also, there is no need to use reference material in order to identify the components because their signals have the same chemical shift even if the matrix is different.
- Sample collection: How did the authors select the sample with similar maturity stages? Then 100 grams from each fruit were used. How did the authors collect the 100-gram samples? From the same part of each fruit or from several parts of the fruits? Please mention it in more detail.
Author’s response: In section 2.2 Sample collection and pre-treatment, between lines 119-124 we added detailed information regarding sample preparation and collection. The authors also thank for these specific questions that lead to an easier understanding of this section.
- Figure 4. Please revise. Remove the 2D plot (left part). It is not in line with the explanation in the text.
Author’s response: We would like to keep figure 4a because a visible improvement of the separation was observed in the case of using a 3D graphic representation compared to the usual ones (2D) and we can suggest that the separation between the sour cherry and cherry group was due to the third discriminant function that was mainly correlated with the sucrose signals, those not being identified in the cherries and sour cherries spectrum or being present in very low concentrations.
- Figure 5 (b). Some samples overlapped. How did you calculate 100% correct classification? Show the result of the classification in detail.
Author’s response: We verified the results and then modified the text between lines 343 – 349.
from \ to |
2015 |
2016 |
2017 |
Total |
% Correct |
2015 |
18 |
0 |
1 |
19 |
94.74% |
2016 |
0 |
36 |
0 |
36 |
100.00% |
2017 |
2 |
0 |
19 |
21 |
90.48% |
Total |
20 |
36 |
20 |
76 |
96.05% |
- The authors mention the use of PCA and LDA. The two are different in the number of samples used in the calculation. For LDA, how did the authors separate the samples (for calibration and prediction)?
Author’s response: The XLSTAT software allow us to select a validation option by using as a validation set a random number of samples defined by the user (24 in our case) and after testing that the results were adequate (obtaining more than 80% for the confusion matrix for the validation sample) all the results were processed taking into account that the model is built on authentic samples.

Round 2
Reviewer 3 Report
Dear authors,
I appreciate your effort to revise the manuscript according to the review.